# The SUMO Pathway in Hematomalignancies and Their Response to Therapies

**DOI:** 10.3390/ijms20163895

**Published:** 2019-08-09

**Authors:** Mathias Boulanger, Rosa Paolillo, Marc Piechaczyk, Guillaume Bossis

**Affiliations:** Equipe Labellisée Ligue Contre le Cancer, Institut de Génétique Moléculaire de Montpellier, University of Montpellier, CNRS, 34000 Montpellier, France

**Keywords:** SUMOylation, leukemia, hematomalignancies, resistance

## Abstract

SUMO (Small Ubiquitin-related MOdifier) is a post-translational modifier of the ubiquitin family controlling the function and fate of thousands of proteins. SUMOylation is deregulated in various hematological malignancies, where it participates in both tumorigenesis and cancer cell response to therapies. This is the case for Acute Promyelocytic Leukemias (APL) where SUMOylation, and subsequent destruction, of the PML-RARα fusion oncoprotein are triggered by arsenic trioxide, which is used as front-line therapy in combination with retinoic acid to cure APL patients. A similar arsenic-induced SUMO-dependent degradation was also documented for Tax, a human T-cell lymphotropic virus type I (HTLV1) viral protein implicated in Adult T-cell Leukemogenesis. SUMOylation also participates in Acute Myeloid Leukemia (AML) response to both chemo- and differentiation therapies, in particular through its ability to regulate gene expression. In Multiple Myeloma, many enzymes of the SUMO pathway are overexpressed and their high expression correlates with lower response to melphalan-based chemotherapies. B-cell lymphomas overexpressing the c-Myc oncogene also overexpress most components of the SUMO pathway and are highly sensitive to SUMOylation inhibition. Targeting the SUMO pathway with recently discovered pharmacological inhibitors, alone or in combination with current therapies, might therefore constitute a powerful strategy to improve the treatment of these cancers.

## 1. Introduction

In mammals, SUMO is a family of related ubiquitin-like peptidic post-translational modifiers (collectively called SUMO hereafter), of which SUMO-1 to -3 are the best-studied members. SUMO-1 shares around 50% identity with SUMO-2 and -3, which are 97% identical. Experimentally, SUMO-2 and -3 are most often indistinguishable and will be referred to as SUMO-2/3 below. Although showing functional specificities, SUMO-2/3 can nevertheless compensate for the knockdown of SUMO-1 in mice [1]. Other less characterized SUMO isoforms were also described. These are SUMO-4, the polymorphism of which has been linked to type 2 diabetes [2], and SUMO-5, which seems to be expressed specifically in certain tissues such as testes and the hematopoietic system [3]. However, their expression and conjugation at endogenous level is still debated.

SUMOylation consists of the covalent C-terminal conjugation of SUMO to the ε-amino group of lysines from target proteins via an isopeptide bond. This process is reminiscent of protein ubiquitylation but involves dedicated enzymes [4]. SUMO is, first, activated via covalent attachment to the reactive cysteine of a heterodimeric E1 SUMO-activating enzyme (Aos1/Sae1-Uba2/Sae2, Sae standing for SUMO-activating enzyme) and, then, transferred to the reactive cysteine of a E2 SUMO-conjugating enzyme (Ubc9/Ube2i). Finally, E3 factors facilitate the transfer of SUMO from the E2 to lysines of target proteins. Only a few E3s have been characterized so far. Among them, the nucleoporin RanBP2 is localized on the outer side of the nuclear pore complex and functions as an allosteric activator of Ubc9 [5]. The E3s of the PIAS family, which all contain a SP-RING domain, display no intrinsic catalytic activity but facilitate the binding of Ubc9 to its substrates. Other ligases, such as ZNF451, can elongate SUMO-2/3 chains [6,7]. Indeed, similarly to ubiquitin, the SUMO proteins can form chains. Thus, SUMO-2 and SUMO-3 can homopolymerize, or heteropolymerize, principally through conjugation to their conserved Lys11. However, chain formation can also implicate other internal Lys residues. Furthermore, SUMO-2/3 can form mixed chains with SUMO-1. However, the latter acts as a chain terminator, as none of its lysine can be conjugated by any of the SUMO, including itself (Figure 1).

Importantly, most SUMOylated proteins go through constant cycles of conjugation/deconjugation due to a variety of cell deSUMOylases that can cleave the isopeptide bonds involving SUMO. These include the isopeptidases from the SENP family (SENP-1, -2, -3, -5, -6, -7), which have different intracellular localizations and SUMO paralogue specificities [8], as well as USPL1, which localizes to Cajal Bodies [9,10], and DeSi-1 and -2 [11], which have not been fully characterized yet.

As for most other post-translational modification, SUMOylation changes target protein interaction surfaces. It either masks certain domains, induces intraprotein conformational changes or brings a new peptidic moiety, i.e., SUMO itself, which can recruit proteins harboring so-called SUMO-Interacting Motifs (SIM). Of note, SIMs are short motifs with a hydrophobic core that are found in many polypeptides [12]. 

Recent high throughput mass-spectrometry-based experiments have revealed that more than 6000 proteins can be SUMOylated [13,14], which is, most probably, an underestimation of the actual number of SUMOylatable substrates within cells. Consistently, SUMOylation has been implicated in the control of numerous cellular and physiological processes [15]. Many components of the SUMO pathway are indeed essential for life in various organisms. For example, inactivating Ubc9 in mice leads to early embryonic death due to major mitotic defects [16]. As SUMO-conjugating- and -deconjugating machineries are principally nuclear, SUMOylation occurs mostly in the nucleus. Yet, various cytosolic- or plasma-membrane-associated processes were also found to be controlled by SUMOylation [17]. One particularly well-studied example of nuclear SUMOylation is DNA repair [18]. Upon DNA double-strand break formation, many proteins get SUMOylated at lesion sites, in particular thanks to the recruitment of PIAS E3s [19]. This entails a local wave of SUMOylation, which stabilizes protein/protein interactions within the supramolecular complexes responsible for DNA repair [20]. Interestingly, SUMOylation at DNA breaks subsequently triggers the recruitment of SUMO-binding proteins via their SIMs. This is the case of the SUMO-targeted Ubiquitin Ligase (StUBL) RNF4. It binds to SUMOylated MDC1, ubiquitylates it and sends it for degradation, which is required for efficient repair [21]. The other best-studied cellular function that is widely controlled by SUMOylation is transcription [22,23,24]. Indeed, transcription factors, transcriptional regulators, and chromatin remodelers constitute the largest class of SUMOylated proteins. It is now clear that SUMO constitutes a mark found at many places within chromatin with, however, enrichments at gene promoters and enhancers [25,26]. The role of SUMOylation in the regulation of transcription is, nevertheless, still ill-understood and is likely to be several-fold. For example, SUMOylation has initially been implicated in transcriptional repression and chromatin silencing, in particular through the recruitment of Polycomb repressive complexes [26,27]. However, it has also been associated more recently with highly active transcription, such as that of histone and tRNA genes [25], where it, generally, limits expression.

SUMOylation is highly sensitive to various stresses that alter the activity of SUMO pathway enzymes [28,29]. For example, reactive oxygen species (ROS) can inactivate SUMO conjugation by inducing the formation of a reversible disulfide bridge between SAE2/Uba2 and Ubc9 catalytic cysteines [30]. This disrupts the SUMOylation/deSUMOylation cycle, resulting in cell protein deSUMOylation, which is required for survival under oxidative stress [31,32]. Various other stresses, such as proteotoxic ones, have also been shown to affect SUMOylation [28]. Heat shock was for instance reported to induce global protein SUMOylation, which protects a number of protein complexes, notably within chromatin [33]. 

### Implication of SUMOylation in Carcinogenesis

As many enzymes are involved in the SUMO conjugation/deconjugation cycle and as the number of SUMO targets and regulators is high, dysregulations of the SUMO system are expected to impact cellular behavior. This could in turn facilitate the onset and progression of various human diseases, in particular cancer [34]. Indeed, SUMOylation enzyme abundance and/or activity have long been shown essential for tumorigenesis in various solid tumors. For example, RNAi-directed depletion of SAE1 and SAE2/Uba2 [35], as well as that of Ubc9 [36], are synthetically lethal with oncogenically mutated K-Ras. Similarly, SAE2/Uba2 is synthetically lethal with the Myc oncogene when overexpressed in aggressive breast cancers [37]. Growth of Notch1-driven breast epithelial cancer cells is also dependent on an active SUMO pathway as well [38]. Moreover, Ubc9 was found overexpressed in a number of solid tumors (ovary, colon, prostate, melanoma, lung, glioma, squamous cell carcinoma), most often aggressive, which is associated with higher SUMOylation activity. SUMO E3s, such as PIAS1, PIAS3, PIAS4, RanBP2, Pc2, or PML, were also found expressed at higher levels in various tumors (colon, stomach, prostate, lung, brain, breast, liver etc.) as compared to normal controls and, sometimes, shown instrumental for the tumor phenotype. Interestingly, RanBP2/Nup358 is engaged in fusion proteins resulting from chromosomal translocations found in several hematomalignancies. For example, a t(2:8) translocation fuses the N-terminal moiety of RanBP2/Nup358 to the C-terminal moiety of the tyrosine kinase receptor FGFR1 in a myeloproliferative/myelodysplastic neoplasm [39] and a inv(2)(p23q21) translocation fuses the same N-terminal domain of RanBP2/Nup358 to the C-terminal moiety of the tyrosine kinase ALK in both a large B-cell lymphoma [40] and an acute myeloid leukemia [41,42,43]. For more details on the deregulations of SUMO enzymes in cancer, the reader is referred to recent reviews [34,44,45]. Finally, spontaneous or germline mutations in SUMOylation sites of critical cell fate regulators can also contribute to carcinogenesis. For example, a germline mutation entailing loss of SUMOylation of the transcription factor MITF predisposes patients to melanoma and renal carcinoma development [46,47]. Along the same line, a missense mutation in the FANCA protein identified in a breast tumor was shown responsible for increased FANCA SUMOylation. This triggers its RNF4-dependent ubiquitylation and degradation by the proteasome and, thereby, contributes to Fanconi anemia DNA repair pathway alteration [48]. 

On their own, these non-exhaustive data indicate that dysregulations of the SUMO pathway play essential roles in both tumorigenesis and response to therapies in different solid cancers. Hematological malignancies do not depart from them. They are reviewed hereafter in a context where promising pharmacological inhibitors targeting SUMOylation have recently been discovered and open novel therapeutic perspectives. 

## 2. SUMOylation in Acute Myeloid Leukemia

Acute Myeloid Leukemia (AML) are a heterogenous group of severe hematological malignancies induced by oncogenic transformation of hematopoietic stem cells and myeloid progenitors. In developed countries, AML incidence and mortality rates are 5–8- and 4–6/100,000 per year, respectively [49]. Although new promising molecules have recently emerged (inhibitors of FLT3, IDH1/2, Bcl2, etc.), the standard treatment of AMLs still largely relies on genotoxic-based chemotherapies combining anthracyclines (daunorubicine or idarubicine) and a nucleoside analogue (cytarabine) [50]. However, their prognosis is dismal with 5-year survival rates around 40% in younger patients and much lower in the elderly. One exception concerns the Acute Promyelocytic Leukemia (APL) type of AMLs, which is efficiently cured using a differentiation therapy. It combines all-trans-retinoic acid (ATRA) and arsenic trioxide (As_2_O_3_ ), which leads to the degradation of the oncogene and restores the differentiation of the leukemic cells (see below) [51]. Evidence has accumulated that, on the one hand, an active SUMOylation pathway is crucial for successful treatment of APLs and, on the other hand, it might be exploited for therapeutic purposes in the case of non-APL AMLs.

### 2.1. SUMO in Acute Promyelocytic Leukemias and Their Response to As_2_O_3_-Based-Therapies

APL is a rare (around 10% of AMLs), though extremely malignant, disease because of its very fast spontaneous evolution and occurrence of sudden hemorrhages mainly caused by coagulation defects. It is associated with specific chromosomal translocations that always involve the retinoic acid receptor α (RARα) gene (RARA) on chromosome 17 to create a variety of oncogenic RARα fusion proteins. The most common (>98%) translocation in APLs is t(15;17), which gives rise to a PML/RARα chimeric protein [52]. Importantly, PML-RARα exerts a dual dominant-negative activity on the protein products of natural, non-translocated PML and RARA genes. On the one hand, it represses RARα signaling and, on the other, it disrupts intranuclear domains known as PML nuclear bodies (NBs) [53,54]. PML, which is one of the first characterized SUMOylated proteins [55], constitutes the outer shell of the NB spheres and is the organizer of these domains spread out in the nucleus. SUMOylation of PML is dispensable for NB formation but critical for the recruitment of multiple proteins in NBs via SUMO-SIM interactions [56]. In APLs, the fusion protein PML/RARα leads to the disruption of PML-NBs and the inhibition of RARα-dependent transcriptional programs involved in differentiation. As_2_O_3_ used to treat APL binds directly to PML/RARα and PML and triggers their polymerization via the oxidation of specific cysteines and the formation of disulfide bonds [57]. Consequently, PML aggregates at the outer shell of NBs and gets massively SUMOylated. HyperSUMOylated PML/RARα recruits the SUMO-dependent ubiquitin ligase RNF4, which ubiquitylates its PML moiety, allowing its recruitment to the proteasome and, ultimately, the degradation of the whole oncogenic fusion protein [58,59]. This degradation allows for reactivation of RARα signaling and reformation of NBs, as well as activation of the p53 pathway and, thereby, apoptosis of leukemic cells [60] (Figure 2).

### 2.2. SUMOylation in Non-APL Acute Myeloid Leukemia

Different studies have addressed the role of SUMOylation of specific proteins playing key roles in leukemogenesis and/or the response to therapies of non-APL AMLs. A first example is that of C/EBPα, which is a critical regulator of early myeloid differentiation that is mutated in approximately 10% of AMLs. Depending on their mutations, a mutated gene produces either a transcriptionally inactive protein or a truncated p30 variant, both acting as a dominant-negative effector neutralizing the tumor suppressor activity of the full size p42 C/EBPα isoform [61]. Overproduction of p30 C/EBPα was shown to lead to an increase in Ubc9 gene expression [62]. Moreover, increased Ubc9 activity is responsible for inactivating p42 C/EBPα through its SUMOylation, which limits its pro-differentiation potential and makes the disease phenotype more aggressive [62,63]. SUMOylation was also found to strongly affect IGF-1R (insulin growth factor-like receptor 1) protein activity, which can be upregulated in AMLs. Interestingly, AML cell proliferation could be decreased either upon inhibition of Ubc9 or mutation of the IGF-1R SUMOylation sites without, however, altering cell apoptosis [64]. PRDM16 is a transcriptional repressor and overexpression of one of its isoforms, sPRDM16, is oncogenic in leukemia [65]. sPRDM16 was shown to be SUMOylated on its lysine 568. This SUMOylation is required to fully repress transcription, promote proliferation, and inhibit differentiation of AML cells both in vitro and in mouse xenografts of human AML cells [66,67]. HIPK2 is a kinase, which is part of the AML1 complex and participates in its activation. Mutations of HIPK2 (R868W and N958I) impairing its ability to activate AML1 have been identified in both AML- and Myelodysplastic Syndrome (MDS)-suffering patients [68]. Interestingly, these mutations affect the ability of HIPK2 to interact non-covalently with SUMO via its SIM and, as a consequence, inhibit its SUMOylation and recruitment to PML-NBs [69]. Another important role for SIMs was demonstrated for c-Myb, a transcription factor whose dysregulation is also involved in leukemogenesis [70]. Thus, mutation of its SIM increases c-Myb transcriptional activity and potentiates its oncogenic activity in myeloid cells [71].

Besides these observations on specific pro-leukemogenic proteins, the SUMO pathway, taken as a whole, has been implicated in non-APL AML response to treatments. Using AML cell lines and patient samples, recent work by our group showed that the chemotherapeutics (Daunorubicin, Cytarabine, and Etoposide) used in standard treatment of non-APL AMLs induce a rapid and massive intracellular protein deSUMOylation in chemosensitive cells [72]. Such a deSUMOylation starts, at moderate levels, earlier than caspase activation, i.e., the irreversible phase of apoptosis, and becomes massive when cells progress towards death. This results from direct inhibition of the E1 and E2 SUMOylation enzymes by chemotherapeutics-induced ROS, which promote the formation of a disulfide bridge between the catalytic cysteines of the two enzymes [30]. ROS-dependent deSUMOylation participates in the induction of the pro-apoptotic gene ddit3 (also called CHOP-10 or GADD153) through deSUMOylation of proteins bound to its transcription promoter region (Figure 3). By contrast, in chemoresistant cells, the chemotherapeutics do not induce the ROS/SUMO axis. However, the latter could be reactivated by pro-oxidants or by inhibition of the SUMO pathway using either a pharmacological inhibitor of SUMOylation (anacardic acid) or RNA interference targeting the different SUMO isoforms, which leads to tumor cell death. Interestingly, anacardic acid efficiently kills patient leukemic stem cells (LSCs; defined as CD34^+^ CD38^low/-^CD123 ^+^ cells) and limits growth of a chemoresistant human AML cell line xenografted to immunodeficient mice [72]. These observations were recently strengthened using another pharmacological inhibitor of the SUMO pathway (2-D08) that targets the SUMO E2 enzyme [73]. 2-D08 was shown to induce apoptosis of various AML cell lines through ROS production, possibly via the deSUMOylation of the NADPH oxidase Nox2 [74]. Altogether, these data suggest that targeting the SUMO pathway might be exploited to overcome chemoresistance in non-APL AMLs.

SUMOylation has more recently been associated with leukemic cell response to epigenetic drugs such as HDAC inhibitors (HDACi). In particular, the pan-HDACi SAHA was shown to induce the SUMO-2/3 modification of the polycomb complex protein CBX2. This leads to RNF4 recruitment and the proteasome degradation of CBX2, which results in reduced proliferation of the leukemic cells [75]. 

SUMOylation has also been involved in non-APL AMLs response to differentiation therapies. As already mentioned, differentiation therapies using ATRA are efficient at curing APLs. However, even though ATRA has been reported to induce (at least to some extent) the differentiation of other AML subtypes in vitro, clinical trials have always failed to show significant efficacy of ATRA-based differentiation therapies in non-APL AMLs [76]. SUMOylation of RARα on its lysine 339 was formerly reported to increase its stability [77]. Moreover, the same authors also showed that inhibition of SUMOylation was limiting ATRA-induced differentiation of AML cell lines [77]. Contrasting with these results, we recently reported that both pharmacological and genetic inhibition of SUMOylation promote, not only the differentiation, but also the inhibition of proliferation of ATRA-treated non-APL AML cell lines and -primary patient samples [78]. Moreover, we also showed that SUMOylation inhibition, at the level of the chromatin, facilitates the induction of genes involved in differentiation, cell cycle arrest, and apoptosis [78] (Figure 3). Interestingly and potentially explaining the differences in outcomes between the two above-described studies, fine-tunable RNAi against Ubc9 recently showed that low to mild inhibition of SUMOylation increases ATRA-induced expression of differentiation markers, whereas strong inhibition has the opposite effect. However, in all cases, inhibition of SUMOylation strongly increased the anti-proliferative activity of ATRA [26]. Altogether, these data suggest that inhibition of SUMOylation might help improve the anti-leukemic activity of ATRA in non-APL AMLs.

## 3. The SUMOylation Pathway in Multiple Myeloma

Multiple myeloma (MM) is the second most common hematological malignancy. It accounts for >10% of blood cancers and approximately 1% of all cancer types. It is characterized by the clonal proliferation of malignant plasma cells within the bone marrow, lytic bone lesions, and immunodeficiency. It is also associated with the production of high levels of a monoclonal protein (immunoglobulin or component of immunoglobulin) in the blood and/or urine [79,80]. Despite these common features, MM is a heterogeneous disease involving a variety of complex oncogenic molecular events. As a consequence, patient survival remains highly variable and cannot be accurately predicted with current models, as the cellular pathways that determine patients’ responses to treatments remain largely unidentified. The survival of patients has significantly improved over the past 20 years owing to the advent of proteasome inhibitors (such as bortezomib or carfilzomib, others currently being developed) and immunomodulatory drugs (such as thalidomide or lenalidomide) combined with other chemicals (such as melphalan, cyclophosphamide, doxorubicine, busulphan, etoposide, cis-platin, vincristine, prednisone, or dexamethasone), autologous stem cell transplantation, radiotherapy, and monoclonal antibody treatment. MM remains however largely fatal as most patients are either refractory to treatment or relapse after acquisition of chemoresistance [81,82]. New therapeutic approaches are therefore urgently needed to improve patients’ outcomes. In this prospect, there is recent evidence that targeting the SUMOylation pathway may open a novel therapeutic window.

In a pioneering work, it was reported that the SUMOylation pathway is often overactivated in MM and associated with adverse patient outcomes [83]. Using MM bone marrow samples from newly diagnosed, previously untreated patients, they first showed that SUMOylation was markedly enhanced in MM patients as compared to normal B- and plasma cells from healthy individuals. Similar results were obtained when comparing myeloma cell lines to normal B- and plasma cells. Interestingly, the SUMO-conjugating enzyme Ubc9, the SUMO-E3 PIAS1 and the SUMOylation-inducer tumor suppressor ARF were found more elevated in MM patient samples and cell lines than in controls. Moreover, they appeared to constitute early events of myelomagenesis. Strikingly, 80% of melphalan-based high-dose chemotherapy-treated (a standard treatment at the time the study was conducted) patients with both low (below the median) Ubc9 and low PIAS1 were living 6 years after transplantation, whereas only 45% of patients with high expression survived 6 years, suggesting adverse effects of enhanced SUMOylation activity in MM patients. A number of in vitro observations further supported this idea. In particular, expression of an Ubc9 dominant-negative mutant in MM cell lines γ-irradiated to induce DNA damage showed decreased survival and enhanced apoptosis. This was consistent with the fact that Ubc9 and PIAS1 are rapidly induced and associate upon γ-irradiation of MM cell lines to increase SUMOylation and suggested a protective effect of increased SUMOylation against DNA-damaging agents. The same dominant-negative mutant decreased both proliferation and adhesion of MM cells to bone marrow stromal cells (BMSCs). These are potentially important results, as adherence to BMSCs is essential for localizing MM tumors within the bone marrow environment and triggering the secretion of IL-6 by BMSCs, this cytokine being a crucial proliferation factor for MM cells.

The above-described data indicate that overexpression of Ubc9 confers MM cells with multiple advantages to promote tumorigenesis and predicts decreased MM patient survival when combined with overexpression of PIAS1. They also suggest that the inhibition of the SUMO pathway may represent a novel therapeutic approach to treat MM. However, the overall picture might be more complex, as it was reported that the deSUMOylase SENP1 is also overexpressed in certain MM cell lines and patient samples. Knocking down SENP1 in MM cell lines inhibited proliferation and increased apoptosis [84]. Interestingly, one of the MM cell lines shown to overexpress SENP1 [84] had previously been shown to also overexpress Ubc9 and PIAS1 [83]. This suggests that SUMOylation must be maintained within a specific window to confer selective advantage to MM cells. Up- or down-modulation of SUMOylation might thus represent a therapeutic strategy to improve MM patients’ outcomes. However, it will be essential to accurately define the level of SUMOylation activity conferring selective advantage to MM cells without inhibiting proliferation and inducing cell death. It will also be important to identify the mechanisms whereby transcription of SUMOylation enzyme genes is stimulated. For example, it was suggested that enhanced SENP1 expression in MM cells might be dependent on IL-6 produced by BMSCs [84]. Additionally, the specific proteins, the SUMOylation of which confers pro-tumorigenic properties to MM cells (whether those are increased proliferation or adherence to BMSCs or resistance to currently used treatments) will have to be characterized. Along this line, it is worth noting that β-catenin SUMOylation has been suggested to be involved in the deregulated proliferation of MM cells [85]. SUMOylation inhibition down-regulates the Wnt/β-catenin pathway by promoting ubiquitin-proteasome degradation of β-catenin [85]. Altogether, these data suggest that SUMOylation pathway is overexpressed in Multiple Myeloma and associated with poor prognosis. This makes it an attractive target to improve its treatment. 

## 4. SUMO in B Cell Lymphoma: Therapeutic Vulnerabilities of Myc-Overexpressing Cells

B cell lymphomas are a group of hematomalignancies affecting B cells. They include both Hodgkin’s and non-Hodgkin lymphomas, the latter being the most common group of B cell lymphomas (e.g., follicular lymphomas, mantle cell lymphomas, marginal zone B-cell lymphoma and Diffuse large B-cell lymphomas (DLBCL), Burkitt Lymphoma) [86]. 

The oncogenic Myc protein is associated with aggressiveness of various cancer types and there is ample evidence that its expression is required for maintenance of many B cell lymphoma. It can be overexpressed due to t(8:14) translocations, such as in Burkitt lymphoma, or to amplification and/or post-transcriptional/translational mechanisms in other situations [87]. Myc best documented physiological function is coordination of cell growth, division, and metabolism, which can be deregulated in the case of cancerous overexpression and associated with induction of differentiation blockade and promotion of angiogenesis [88]. 

The first links between SUMO and Myc were obtained in breast cancer model cell lines where overexpression of Myc was shown to be synthetic lethal with the loss of Uba2/Aos1 SUMO E1 and Ubc9 SUMO E2 activity [37]. The same was then shown to be true in Myc-driven B lymphoma [89]. A marked upregulation of genes encoding components and regulators of the SUMO pathway was observed in human B lymphoma cells including Burkitt lymphoma (BLs) and diffuse large B cell lymphoma (DLBCLs) expressing a regulatable Myc gene. This was also the case in primary B lymphoma and in Eμ-Myc transgenic mice expressing Myc under the immunoglobulin heavy chain gene enhancer. Importantly, such an overexpression correlates with increased SUMOylation. The SUMO pathway genes induced include those for E1 (SAE1 and SAE2/Uba2), E2 (Ubc9), certain E3 (Ranbp2, Cbx4, Pias1, Pias2, Pias4), and SUMO-1 to -3. Noteworthy, SUMO-2/3 overexpression was seen in all BLs but was less prevalent in other lymphomas whereas SUMO-1 overexpression was found in all lymphomas. Certain of the SUMO pathway genes are most probably direct targets of Myc, though more functional/molecular analyses are still required to establish this point firmly. Worth noting, the expression of several SUMO pathway genes in various subsets of mouse mature B cells and immature B progenitors was concordant with that of Myc at various stages of B-cell maturation. This suggests that the Myc-SUMO axis is operational, not only in cancer-, but also in normal cells. Moreover, RNAi knock-down of Aos1 and SAE2 suppressed Myc-induced lymphoma cell proliferation via disruption of the G2/M transition and pharmacological inhibition of E1 and E2 activity by anacardic acid and 2-D08, respectively, led to synthetic lethality with Myc overexpression. Finally, RNAi-mediated suppression of SAE2 impaired the development of mouse and human lymphoma after grafting to isogenic or immunocompromized mice, respectively. Altogether, these data indicate non-oncogene addiction for the SUMO pathway of Myc-driven lymphoma and point to the SUMO pathway as a novel potential pharmacological target to treat this cancer [89] (Figure 4). 

Interestingly, Myc itself is a SUMO substrate [90,91,92], with the E3 PIAS1 stimulating its SUMOylation in lymphoma cells and being essential to maintain Myc oncogenic activity in this cancer type [91,92]. A significant percentage of DLBCLs cells where Myc is deregulated were also found positive for PIAS1 whereas healthy lymphoid tissues and resting B cells were essentially negative for both proteins. PIAS1 was shown to promote the ability of Myc to stimulate cell proliferation and survival in Myc-driven B cell lymphoma as well as to form tumors in vivo, as assayed in xenografting experiments [92]. The consequences of Myc SUMOylation are however conflicting since one study suggests that it facilitates its degradation via the recruitment of the RNF4 StUbL, in osteosarcoma epithelial cell context [91] and another that Myc SUMOylation leads to its stabilization and enhances its transcriptional activity, in particular by increasing its interaction with its transcriptional partner Max [92]. 

In conclusion, the SUMO pathway has strong links with the Myc oncogene, which is a major oncogene in Lymphomas. The addiction of Myc-overexpressing lymphomas to SUMOylation makes them particularly good candidates for molecules targeting this pathway (see Section 6).

## 5. SUMOylation of Human T-Cell Lymphotropic Virus Type I (HTLV1) Protein Tax in Acute T-Cell Leukemia/Lymphoma (ATL) 

Adult T-cell leukemia/lymphoma (ATL) is an aggressive malignancy associated with chronic infection by the human T-cell lymphotropic virus type I (HTLV-1), which leads to transformation of CD4^+^ T cells. ATLs typically develop after long latency periods in 3%–5% of the 10–20 million infected individuals in the world. Like many other hematomalignancies, dismal outcomes and relapses are due to intrinsic chemotherapy resistance [93].

After infection, HTLV-1 integrates into the genome of infected cells and the virally-encoded transactivator protein Tax is expressed and interferes with different cellular processes including proliferation, apoptosis, T-cell activation, transcriptional and epigenetic programs, and DNA repair [94]. In particular, disruption of cellular DNA damage response creates genetic instability, which is essential for transformation and progression towards leukemia/lymphoma. Although the role of Tax in leukemogenesis initiation is widely accepted, its role as an ATL driver operating at later stages has long been a matter of debate [93]. This is, in particular, due to the discovery of HBZ, a transcription factor encoded by the virus, which could be the main actor in HTLV transforming potential [95,96,97]. Recent work has, however, suggested that, at least in a number of ATL lines, HTLV-1-transformed cells are addicted to Tax expression for their survival [98]. This suggests that targeting Tax for degradation might represent a therapeutic option to get rid of leukemic cells in a fraction of ATL patients. 

Interestingly, the combination of arsenic trioxide and interferon alpha (IFNα) exerts strong antiproliferative/proapoptotic activities in human ATL patient-derived cell lines and in murine ATLs derived from Tax transgenic mice [99,100]. Moreover, the combination of arsenic trioxide, IFNα, and the nucleotide analog zidovudine could induce complete and durable remissions in human patients [101]. The situation of Tax in arsenic trioxide- and IFNα-treated cells is reminiscent to that of PML/RARα in APLs (see above) (Figure 5). Indeed, in As_2_O_3_/IFNα-treated HTLV-1 transformed- or ATL patient cells, Tax is recruited to NBs, most probably owing to its four SIM domains. Then, it undergoes successively (i) PML-dependent polySUMOylation by SUMO-2/3 but not by SUMO-1, (ii) ubiquitylation by RNF4 and (iii), finally, proteasome-dependent degradation [98]. Of note, RNF4-dependent ubiquitylation of Tax has also been observed by others. Yet, in this case, it was shown to enhance its nuclear export and favor the activation of the NFκB pathway [102]. The experiments were, however, mostly conducted in a non-ATL context (HEK 293 cells). It is also of note that, in untreated ATL cells, Tax may be primarily SUMO-1-conjugated in the nucleus, raising the question of the mechanisms of the SUMO-1 towards SUMO-2/3 shift by arsenic trioxide and IFNα. In conclusion, these data lend support to the idea that the As_2_O_3_/IFNα combination might clear a fraction of ATLs through SUMO/ubiquitin-dependent degradation of its Tax driver. 

## 6. Targeting SUMOylation: Clinical Perspectives

As mentioned above, targeting SUMOylation has recently appeared as a promising approach for the treatment of various hematological malignancies. This has been demonstrated, mostly in vitro, using inhibitors of the SUMO pathway such as Anacardic Acid and 2D-08 (Table 1). However, these molecules lack sufficient specificity and efficiency as well as clinically relevant pharmacological properties [103]. An important breakthrough in the field of SUMOylation is the recent discovery of highly selective and efficient inhibitors of the SUMO pathway. This is the case of the ML-792, which forms an adduct with SUMO and blocks the SUMO E1 [104]. This mechanism-based inhibitor is highly selective for SUMOylation and efficient in the nanomolar range. Interestingly, ML-792 inhibits cancer cell lines proliferation in vitro and cells overexpressing the Myc oncogene are more sensitive to ML-792. Surprisingly, ML-792 was shown to have only minor effects on the regulation of gene expression. It was also not inducing DNA-damage or affecting DNA-repair following treatment with genotoxics. However, this inhibitor leads to chromosome-segregation defects, which compromises mitosis and results in proliferation arrest and death of the cells [104]. TAK-981, a derivative of ML-792 is currently being tested in a phase 1 clinical trial in patients with metastatic solid tumors and lymphomas (ClinicalTrials.gov Identifier: NCT03648372). Another recently discovered molecule, COH000, inhibits the SUMO E1 by binding to a cryptic allosteric site [105,106]. Although less potent than ML-792 (µM range), it was shown to decrease Myc expression in lymphoma cell lines in vitro and prevented tumors growth when injected peritumorally in mice xenografted with the HCT116 colon cancer cell line [106]. As SUMOylation is highly regulated by ROS, the SUMO pathway could also be indirectly targeted by drugs that modulate oxidative metabolism. This is in particular the case for Arsenic Trioxide, which is already approved in APL treatment or Fenretinide, a synthetic retinoid used in phase I/II clinical trials for hematological malignancies, which induces the production of ROS and ceramide [107,108]. Finally, it is expected that inhibitors of SUMOylation will synergize with already approved therapies, as shown for ATRA in non-APL AMLs [78]. Considering the critical role of SUMOylation in the regulation of gene expression, inhibitors of the SUMO pathway might synergize with already approved epigenetic drugs targeting Histone deacetylase (HDAC) such as panobinostat or vorinostat or DNA-methyltransferase (DNMT) inhibitors such as vidaza or decitabine [109]. 

Targeting the SUMO pathway, alone or in combination with other drugs, is thus a promising approach in the treatment of hematological malignancies. Clinical trials have just begun and SUMOylation inhibitors will face classical challenges of drug development (biodisponibility, toxicity, patient selection) before benefiting patients. More work is also needed to better understand the functional roles of SUMOylation and the relevance of its targeting in hematological malignancies.

## Figures and Tables

**Figure 1 ijms-20-03895-f001:**
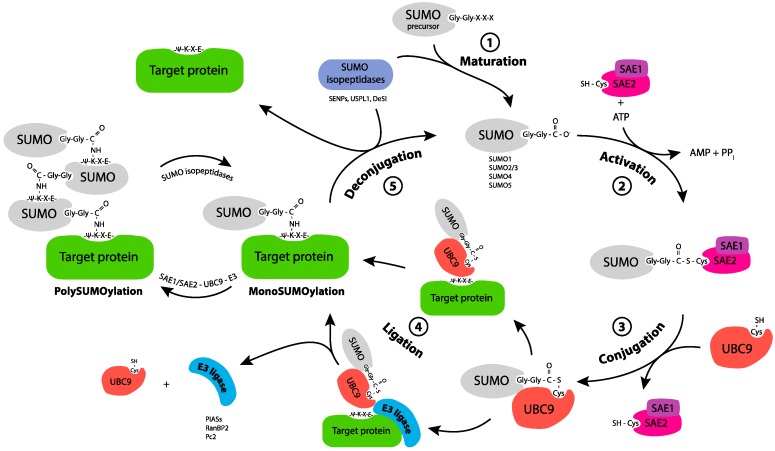
SUMO conjugation/deconjugation cycle. SUMO-1, -2, -3 precursors are maturated through the action of SUMO isopeptidases, which cleave the C-terminus of SUMO to reveal a Gly-Gly motif (1. Maturation). Mature SUMO is activated by forming an ATP-dependent thioester bond between its C terminal Glycine and the catalytic Cys of the E1 enzyme, the heterodimer SAE1/SAE2 (2. Activation). Activated SUMO is transferred, through trans-thiolation, to the catalytic cysteine of the E2 enzyme, Ubc9 (3. Conjugation). Ubc9 triggers the enzymatic transfer to the Lys (K) of the target protein, either alone or with the help of a SUMO E3 ligase through the formation of an isopeptide bond (4. Ligation). The target protein can be monoSUMOylated, multi and/or polySUMOylated. SUMO isopeptidases cleave SUMO from its substrate and thereby release free SUMO (5. Deconjugation).

**Figure 2 ijms-20-03895-f002:**
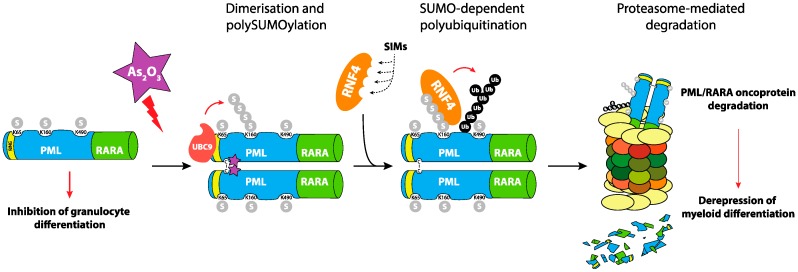
Role of SUMOylation in PML-RARα degradation in Acute Promyelocytic Leukemias (APL). The oncogenic PML-RARα is responsible for the blockage of differentiation of Acute Promyelocytic Leukemic cells. Their treatment with arsenic trioxide leads to the polymerization of the fusion protein via direct binding and reactive oxygen species (ROS)-dependent formation of disulfide bonds. This triggers its poly-SUMOylation and the recruitment of the RNF4 SUMO-targeted Ubiquitin-ligase (StUbL), which ubiquitylates PML-RARα and targets it for proteasomal degradation. This allows the reactivation of the RARα differentiation program, the reformation of PML nuclear bodies (NBs) and the induction of apoptosis of the leukemic cell.

**Figure 3 ijms-20-03895-f003:**
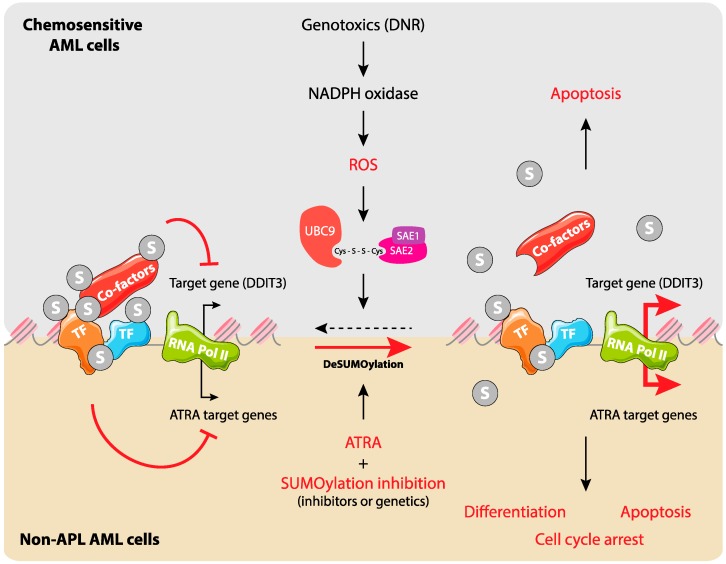
SUMOylation regulates gene expression during Acute Myeloid Leukemia (AML) response to chemo- and differentiation therapies. Upper panel: In chemosensitive AML cells, chemotherapeutic drugs, in particular anthracyclines, induce the production of ROS through the activation of NADPH oxidases. This leads to the inactivation of SUMO E1 and E2 enzymes via the formation of a disulfide bond between their catalytic cysteines. This results in the deSUMOylation of cellular proteins (in priority those bound to chromatin) and participates in the regulation of specific genes such as DDIT3 and entry of cells into apoptosis. Lower panel: in non-promyelocytic AMLs, SUMO participates in silencing of all-trans-retinoic acid (ATRA) target genes. Inhibition of SUMOylation favors ATRA-induced activation of specific genes and leads to cell differentiation, arrest of proliferation, and apoptosis.

**Figure 4 ijms-20-03895-f004:**
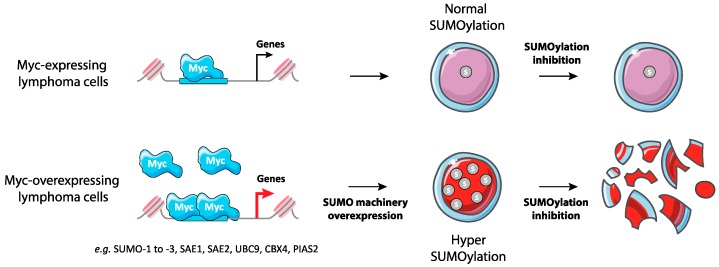
Synthetic lethality between Myc overexpression and inhibition of SUMOylation in lymphoma. Myc-overexpressing B-lymphoma cells show increased expression of many enzymes of SUMOylation enzymes and global hyperSUMOylation. These cells are highly sensitive to the inhibition of SUMOylation.

**Figure 5 ijms-20-03895-f005:**
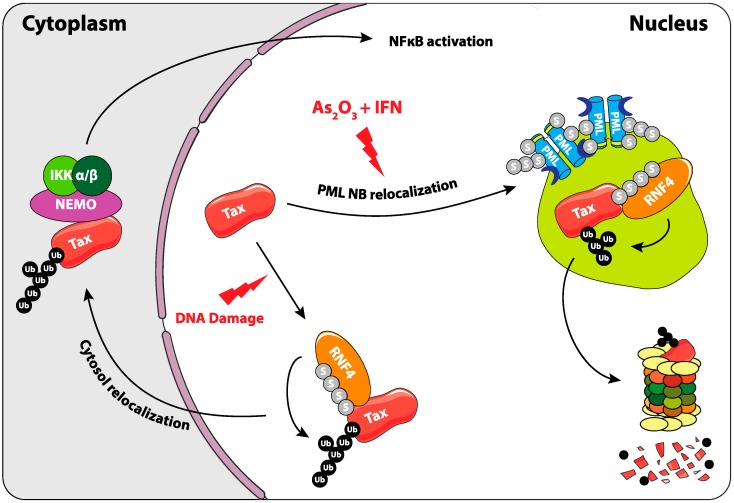
Role of SUMOylation in the regulation of Tax oncoprotein of the human T-cell lymphotropic virus type I (HTLV1) virus. Upon As_2_O_3_ and INFα treatment, Tax is recruited to PML NBs. It is then poly-SUMOylated, recognized by RNF4, poly-ubiquitylated, and degraded by the proteasome. Another involvement of SUMOylation in the regulation of Tax was shown upon DNA damage, where its RNF4-dependant ubiquitylation triggers its cytosolic translocation. This leads to the activation of the NFκB pathway, probably thanks to its ability to bind to and activate NEMO and the IKK complex.

**Table 1 ijms-20-03895-t001:** Inhibitors of SUMO activating and conjugating enzymes. List of the small molecules, which inhibit SUMO conjugation. The IC50 are those obtained with purified recombinant enzymes or on cell lines. Only the molecules that were either validated on cells in vitro or in vivo in mouse models are indicated. Only one is being used in humans (TAK-981).

Inhibitor Name	Structure	IC_50_ (µM)	Target	Use	Reference
Gingkolic acid	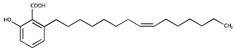	3	SAE1/SAE2	*In vitro*	[72,110]
Anacardic acid	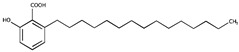	2.2	SAE1/SAE2	*In vitro* *In vivo*	
Kerriamycin B	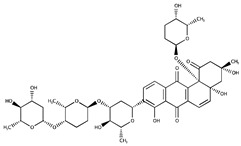	11.70	SAE1/SAE2	*In vitro*	[111]
Spectomycin B1	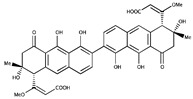	4.4	UBC9	*In vitro*	[112]
2D08	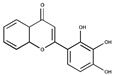	6	UBC9	*In vitro* *In vivo*	[78,113]
Davidiin	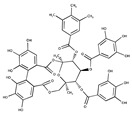	0.150	SAE1/SAE2	*In vitro*	[114]
Tannic acid	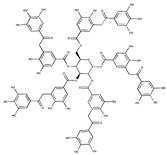	12.80	SAE1/SAE2	*In vitro*	[115]
ML-792	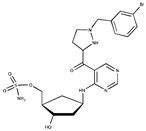	0.003	SAE1/SAE2	*In vitro*	[104]
COH000	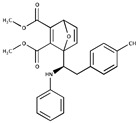	0.200	SAE1/SAE2	*In vitro* *In vivo*	[105,106]
TAK-981	NA	NA	SAE1/SAE2	*In vitro* *In vivo*	ClinicalTrials.gov Identifier: NCT03648372)

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
