# Peer review of "The SUMO Pathway in Hematomalignancies and Their Response to Therapies"

_ijms, 2019, doi:10.3390/ijms20163895_

Round 1
Reviewer 1 Report
This is a comprehensive and well written review which covered majority of aspects on SUMOylation in hematological. Authors, have discussed on why SUMOylation is important and what is the need to target this. One small aspect to improve is to expand on targeting the SUMOylation pathway in hematological malignancies, which will stand out from other reports in literature. Here are my comments/suggestions.
Developing a drug for a new target and preclinical and clinical evaluations take minimum of 10-15 years given the rarity and complexity of some hematological malignancies. Authors have mentioned ROS could be key determinant for SUMOylation process, can authors discuss if ROS inducing drugs that are already in approved/Phase II clinical trails like arsenic trioxide, fenretinide (Pub Med ID: 28420721), etc. could be used as alternative targets for SUMOylation ?
Similarly since HDAC inhibitors have shown promising results in APL and approved in MM could HDAC inhibitors and ROS modulating drugs, used as combination in hematological malignancies to target SUMOylation (example of preclinical activity, Pub Med ID : 28119491)
Since cancer is dysregulated in multiple ways, what would be ideal combinations to target along with SUMOylation and what will be few challenges targeting SUMOylation?
Author Response
We thank all 3 reviewers for their critical insights on our manuscript. We have modified the text and taken all their comments into consideration. Changes in the manuscript are highlighted in red.
Response to reviewer 1
Reviewer 1: Developing a drug for a new target and preclinical and clinical evaluations take minimum of 10-15 years given the rarity and complexity of some hematological malignancies. Authors have mentioned ROS could be key determinant for SUMOylation process, can authors discuss if ROS inducing drugs that are already in approved/Phase II clinical trails like arsenic trioxide, fenretinide (Pub Med ID: 28420721), etc. could be used as alternative targets for SUMOylation ? Similarly since HDAC inhibitors have shown promising results in APL and approved in MM could HDAC inhibitors and ROS modulating drugs, used as combination in hematological malignancies to target SUMOylation (example of preclinical activity, Pub Med ID : 28119491)Since cancer is dysregulated in multiple ways, what would be ideal combinations to target along with SUMOylation and what will be few challenges targeting SUMOylation?
Answer: We have now complemented the part concerning the inhibitors of SUMOylation. We have introduced the idea of indirect targeting through the modulation of ROS. We have also introduce the concept of potential synergism with other drugs, in particular epigenetic therapies.
Reviewer 2 Report
The current review addresses an important issue regarding the deregulation of Sumoylation pathway and its functional implication in hematological malignancies.
The authors, here, provides an exhaustive overview of the altered status of SUMO pathways in hemato-malignancies focusing on molecular targets and molecular pathways through which deregulated-SUMO pathway is involved in tumorigenesis and cell-therapy response.
Minor point.
I would only suggest to the authors to complete the review with a table showing the most recently and effective small-molecule inhibitors targeting the SUMOylation modification pathway, their possible status in the clinical phase study in both hematological and solid tumors.
Author Response
We thank all 3 reviewers for their critical insights on our manuscript. We have modified the text and taken all their comments into consideration. Changes in the manuscript are highlighted in red.
Reviewer 2: I would only suggest to the authors to complete the review with a table showing the most recently and effective small-molecule inhibitors targeting the SUMOylation modification pathway, their possible status in the clinical phase study in both hematological and solid tumors.
Answer: We have now added a table with all known inhibitors of the SUMO pathway
Reviewer 3 Report
This is a very interesting review on the role of SUMOylation in different hematomalignancies. Overall, it is well written and I support its publication after improving the manuscript as described hereafter.
Major points
1-Overall, the flow of the review is very good, with a proper introduction, a general part on SUMOylation in carcinogenesis and then focusing on different hematomalignancies. In contrast, section 6 instead focusses on RanBP2. This disrupts the flow of the manuscript. I would recommend to remove it and instead add 4-5 short lines on the translocations involving RanBP2 in the introduction at the point where RanBP2 is mentioned (line 118). Furthermore, I would recommend removing the sentence on RanBP2 at the end of the abstract (lines 22-23).
2-It is still unclear whether SUMO4 and SUMO5 are properly expressed at the endogenous protein level and conjugated to target proteins (lines 34-36). This section needs to be written more carefully.
3-Most sections end with a nice conclusion; this is very helpful. But it is not done completely systematically – please provide these conclusions at the end of each main section, including for section 3 and 4.
4-The final section, 7, is too brief. More mechanistic details on how the SUMO inhibitor kills cancer cells could be added here.
5-Figure 2: add an extra step to show that As2O3 directly binds to PML.
6-Figure 4: this figure is too simplistic. More mechanistic details on the interplay between SUMO and Myc could be added here.
7-Section 5 would benefit from adding a figure on Tax.
Minor points
8-Several sentences are rather long. For clarity, I would recommend to shorten these sentences.
9-“participates in” instead of “participates on” (abstract).
10-“heteropolymerize” instead of “heteropolymerize between them” (lines 48-49).
11-providing a few mechanistic sentences on the differentiation therapy (line 143) would be interesting for the readers.
12-line 417: “has” instead of “have”.
Author Response
We thank all 3 reviewers for their critical insights on our manuscript. We have modified the text and taken all their comments into consideration. Changes in the manuscript are highlighted in red.
Reviewer 3: Overall, the flow of the review is very good, with a proper introduction, a general part on SUMOylation in carcinogenesis and then focusing on different hematomalignancies. In contrast, section 6 instead focusses on RanBP2. This disrupts the flow of the manuscript. I would recommend to remove it and instead add 4-5 short lines on the translocations involving RanBP2 in the introduction at the point where RanBP2 is mentioned (line 118). Furthermore, I would recommend removing the sentence on RanBP2 at the end of the abstract (lines 22-23).
Answer: we agree with this reviewer and have removed this part. We have introduced RanBP2 translocations in the introduction (lines 121-126).
Reviewer 3: It is still unclear whether SUMO4 and SUMO5 are properly expressed at the endogenous protein level and conjugated to target proteins (lines 34-36). This section needs to be written more carefully.
Answer: A new sentence has been added (line 36): “However, their expression and conjugation at endogenous level is still debated.”
Reviewer 3: Most sections end with a nice conclusion; this is very helpful. But it is not done completely systematically – please provide these conclusions at the end of each main section, including for section 3 and 4.
Answer: Conclusions have now been added to both section 3 and 4
Reviewer 3: The final section, 7, is too brief. More mechanistic details on how the SUMO inhibitor kills cancer cells could be added here.
Answer: This section (now section 6) has been expended. We have better explained how the inhibitors work and given more details on their potential synergies with other therapies. We have also added a table with all known SUMOylation inhibitors
Reviewer 3: Figure 2: add an extra step to show that As2O3 directly binds to PML.
Answer: We now show on the figure that AS2O3 can directly bind to PML
Reviewer 3: Figure 4: this figure is too simplistic. More mechanistic details on the interplay between SUMO and Myc could be added here.
Answer: We have now modified the figure to better show how Myc in involved in the overactivation of the SUMO pathway
Reviewer 3: Section 5 would benefit from adding a figure on Tax.
Answer: A new figure has now been added.
Reviewer 3 Minor points
8-Several sentences are rather long. For clarity, I would recommend to shorten these sentences.
9-“participates in” instead of “participates on” (abstract).
10-“heteropolymerize” instead of “heteropolymerize between them” (lines 48-49).
11-providing a few mechanistic sentences on the differentiation therapy (line 143) would be interesting for the readers.
12-line 417: “has” instead of “have”.
Answer: All these points have now been corrected.